# Spontaneous Closure of a Full-Thickness Traumatic Macular Hole in a Paediatric Patient

**DOI:** 10.3390/diagnostics15040400

**Published:** 2025-02-07

**Authors:** Bogumiła Wójcik-Niklewska, Erita Filipek

**Affiliations:** 1Department of Pediatric Ophthalmology, Faculty of Medical Sciences in Katowice, Medical University of Silesia, 40-055 Katowice, Poland; bniklewska@sum.edu.pl; 2Professor Kornel Gibiński University Hospital Center, Medical University of Silesia, 35 Ceglana Street, 40-514 Katowice, Poland

**Keywords:** macular hole, children, trauma

## Abstract

A macular hole is a defect of the neurosensory retina at the fovea. Post-traumatic holes can occur immediately after blunt trauma, causing severe non-penetrating retinal contusion or after sudden detachment of the vitreous from the retina. Post-traumatic macular holes can close spontaneously or may require vitreoretinal surgery. This paper aims to present the case of an 11-year-old boy with a macular hole following a ball injury. The child reported deterioration of visual acuity. Ophthalmic examination, ocular ultrasound, optical coherence tomography (OCT), perimetry, and a pattern visual evoked potential (VEP) test were performed. On the day of injury, the visual acuity of the right eye was 0.04 and intraocular pressure was 28 mmHg; the eyelid skin was reddened, and superficial conjunctival injection was observed. A fundus examination revealed oedema, pre-retinal haemorrhages, and a macular hole; peripheral retinal oedema in the superior temporal quadrant with pre-retinal haemorrhages was also seen. At the follow-up appointment scheduled 5 months following hospital discharge, visual acuity of the right eye was 0.3 and intraocular pressure was 20 mmHg. Follow-up OCT images of the OD macula were comparable to the findings obtained on the day of hospital discharge, i.e., 10 days after blunt trauma to the right eye. The left-eye OCT did not reveal any abnormalities.

**Figure 1 diagnostics-15-00400-f001:**
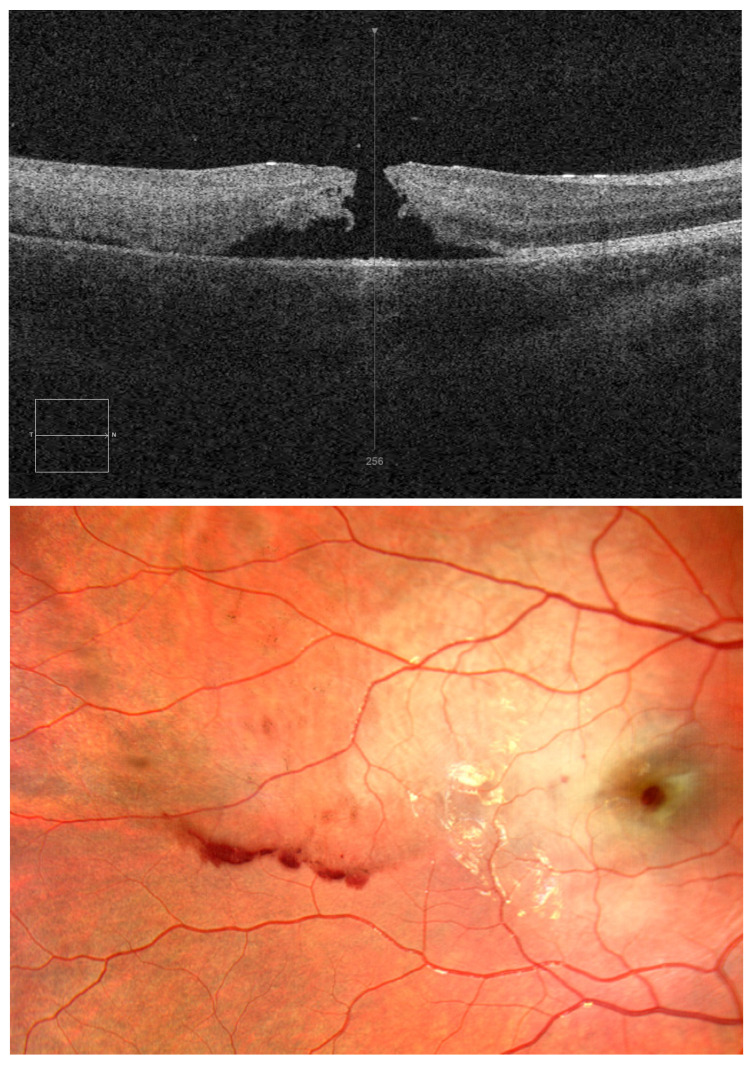
This paper aims to present the case of an 11-year-old boy with a macular hole following a ball injury. We would like to point out the fact that after blunt trauma, a macular hole may appear not only in adults, but also in children. An 11-year-old boy was admitted to the Department of Paediatric Ophthalmology on an emergency basis due to ocular trauma from a football hit to the right eyeball (OD). The child reported pain and deterioration of visual acuity. Ophthalmic examination, ocular ultrasound, optical coherence tomography (OCT), perimetry, and a pattern visual evoked potential (VEP) test were performed. OCT demonstrated a full-thickness macular hole on day 1 after injury. On the day of injury, the visual acuity of the right eye was 0.04 and intraocular pressure was 28 mmHg; the eyelid skin was reddened, and superficial conjunctival injection was observed. A fundus examination revealed oedema, pre-retinal haemorrhages, and a macular hole; peripheral retinal oedema in the superior temporal quadrant with pre-retinal haemorrhages was also seen (Figure 1). OCT showed a full-thickness macular hole in the right eye; no abnormality was found in the left eye. Visual acuity of the left eye was 1.0; intraocular pressure was 18 mmHg. An ocular ultrasound of the right eye depicted single extra echoes in the vitreous chamber; no changes were revealed in the left eye. A macular hole is a defect of the neurosensory retina at the fovea. The most common cause is vitreoretinal traction, which is characteristic of age-related posterior vitreous detachment. However, a macular hole can also result from mechanical ocular blunt injury. Post-traumatic holes can occur immediately after blunt trauma, causing severe non-penetrating retinal contusion, or after sudden detachment of the vitreous from the retina. Post-traumatic macular holes can close spontaneously or may require vitreoretinal surgery. The improvement in visual acuity depends on the extent of the foveal damage [1,2,3,4]. The pattern VEP test revealed the following: PVEP1º: the P100 latency was longer for the right eye; the P100 amplitude was reduced by 50% for the right eye and normal for the left eye; PVEP15’: the P100 latency was slightly longer for the right eye and normal for the left eye, while the amplitude was reduced by 70% and 50% for the right eye and left eye, respectively.

**Figure 2 diagnostics-15-00400-f002:**
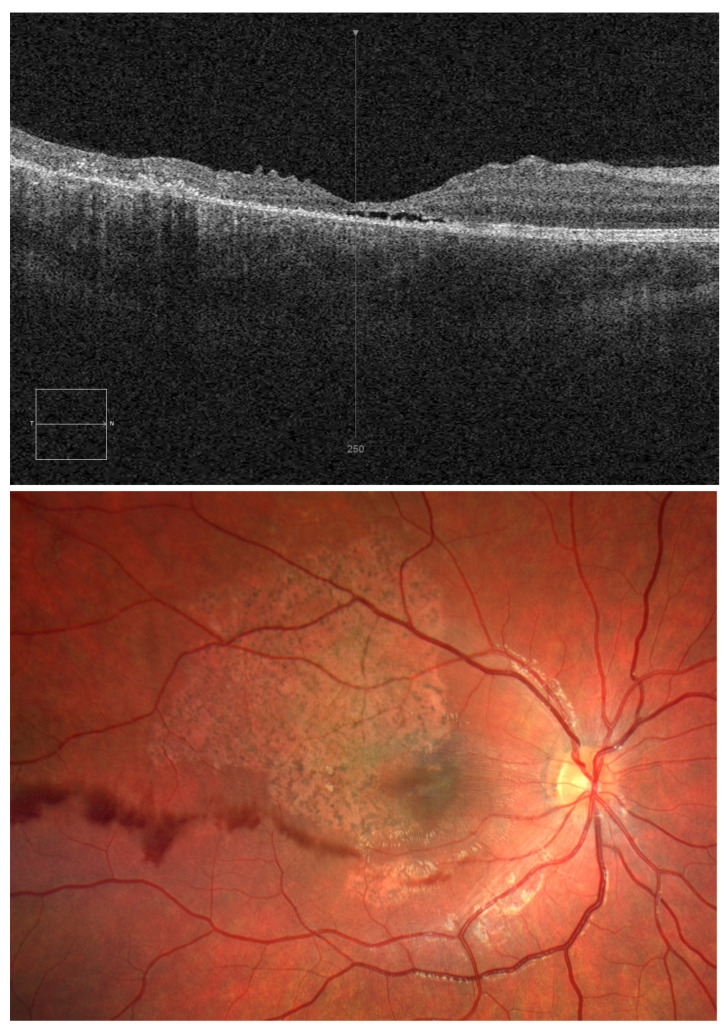
OCT and OD fundus images on day 10 after injury. The child was treated with systemic methylprednisolone, rutoside, and ascorbic acid. Dexamethasone eye drops, ophthalmic diclofenac, dorzolamide, troxerutin, and atropine sulphate were applied to the right eye. After a 10-day hospital stay, visual acuity in the right eye was still 0.04; intraocular pressure was 21 mmHg. OCT revealed closure of the macular hole but disorganised retinal inner layers and a small central fluid space in the macula (Figure 2). The child did not complain of right eyeball pain but remained concerned about poor visual acuity in the right eye.

**Figure 3 diagnostics-15-00400-f003:**
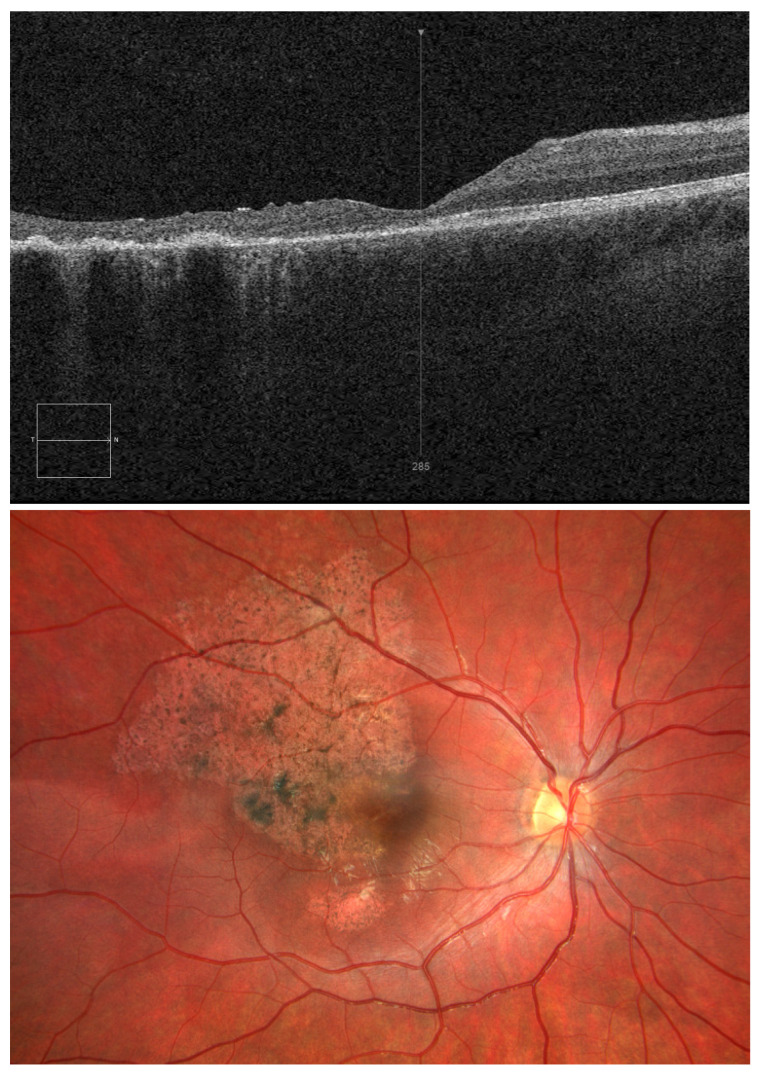
OCT and OD fundus images 5 months after injury. At the follow-up appointment scheduled 5 months following hospital discharge, visual acuity of the right eye was 0.3 and intraocular pressure was 20 mmHg. Follow-up OCT images of the OD macula were comparable to the findings obtained on the day of hospital discharge, i.e., 10 days after blunt trauma to the right eye. Left-eye OCT did not reveal any abnormalities (Figure 3). Given the potential for spontaneous closure of a traumatic macular hole, conservative treatment and OCT monitoring of the lesions may play an essential role in treatment, providing an alternative to surgery [5,6]. Despite the fact that it was spontaneous closure of a traumatic macular hole, there was irreversible damage to the retinal layers in the central fovea, which resulted in poor visual acuity. It is important to evaluate the retina with OCT scans after ocular trauma in children in order to decide on treatment and determine the prognosis for visual acuity. It appears that the retinal changes seen on the fundus photographs in our patient—retinal pallor and pre-retinal haemorrhage—also worsened the visual acuity prognosis.

## Data Availability

All relevant data are within the manuscript.

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
