# Peer review of "Spontaneous Closure of a Full-Thickness Traumatic Macular Hole in a Paediatric Patient"

_diagnostics, 2025, doi:10.3390/diagnostics15040400_

Round 1
Reviewer 1 Report
Comments and Suggestions for Authors
The manuscript is well-structured, and the images are clear and informative. The authors have presented a case of spontaneous closure of a traumatic macular hole, a phenomenon that is known in the literature. Given that this case report is being considered under the "Interesting Images" article type, I believe it is important for the authors to provide justification for its inclusion in this category. Specifically, the authors should elaborate on what makes this case particularly unique or noteworthy.
Does the case provide any novel insight or perspective into the pathophysiology or natural history of traumatic macular holes? Are there any rare features, clinical findings, or imaging characteristics that distinguish this case from previously reported instances? Does this case highlight any clinical or diagnostic challenges, or provide an opportunity for educational value beyond what is already well-documented?
Author Response
Reviewer 1
Thank you very much for your review and valuable comments. We agree with your comment.
This case is particularly unique because we would like to pay attention to the fact that after blunt trauma macular hole may appear not only in adults, but also in children. We noted that spontaneous closure occured after conservative management already on the 10th day after the injury, without to perform a posteriori vitrectomy, which is currently used in the treatment of macular holes.
Despite the fact that it was spontaneous closure of a traumatic macular hole there was irreversible damage retinal layers in the central fovea, which resulted in poor visual acuity. It is important to evaluate the retina with OCT scans after ocular trauma in children in order to decide on treatment and determine the prognosis for visual acuity. In appears that te retinal changes seen on fundus photographs in our patient – retinal pallor and preretinal hemorrage – also worsen visual acuity prognosis.
Thank you very much for your comments so that this information can be included in our manuscript.
Reviewer 2 Report
Comments and Suggestions for Authors
This is a nice example of spontaneous closure of a macular hole.
My only comments are:
could you please NOT use the false colour coded image and instead use the monochromatic OCT output when showing the cross-sectional slices? This would enhance the image quality and also link much better with the functional outcomes.
Also: could you please include, the whole image cube as I would imagine you imaged the entire macula to establish the size of the hole.
Author Response
Reviewer 2
Thank you very much for your review and valuable comments. We agree with your comment.
I will convert color OCT scans to monochromatic.
I used Whole image cube to establish the size of the macular hole. I currently do not have access to this imaging in the OCT device I used to examine our patient earlier. I’m so sorry.
Kind regards
BogumiÅ‚a Wójcik-Niklewska
Round 2
Reviewer 1 Report
Comments and Suggestions for Authors
Thank you to the authors for their detailed response. It is well known that traumatic macular holes can occur following trauma in both children and adults. Management of these traumatic macular holes can be categorized into observation or pars plana vitrectomy, which may be performed either early or late. It is also recognized that these holes can close spontaneously without surgical intervention (https://doi.org/10.1016/j.ajo.2024.05.001).